# Impact of Social Media Behavior on Privacy Information Security Based on Analytic Hierarchy Process

**Yuxuan Liu** [1] **, Woon Kwan Tse** [2,*] **, Pui Yu Kwok** [2] **and Yu Hin Chiu** [2]

[1] School of Economics and Management, Wenzhou University of Technology, Wenzhou 150080, China; yxliu26-c@my.cityu.edu.hk

[2] Department of Information Systems, City University of Hong Kong, Hong Kong 999077, China; puiyukwok6-c@my.cityu.edu.hk (P.Y.K.); yuhinchiu3-c@my.cityu.edu.hk (Y.H.C.)

[*] Correspondence: iswktse@cityu.edu.hk

**Abstract:** In the era of global social media, Internet users' privacy rights have been weakened, and the insight and alertness of individuals for privacy disclosure are decreasing. The security and flexibility of the system are usually the two ends of the measurement standard. While more and more users pursue the intelligence and convenience of using social media applications, letting big data technology and AI algorithms learn and use our privacy data cannot be avoided. On the basis of literature review, this paper summarized four categories of social media user behavior, which were divided into privacy concern behavior, privacy protection behavior, active disclosure behavior, and passive participation behavior. Using analytic hierarchy process, this paper explored their relationship with five different types of privacy: defensive privacy, identity authentication privacy, interactive privacy, psychological privacy, and integration information privacy. We finally formulated the most common twelve kinds of sub-internet user behavior on the degree of personal privacy disclosure, and provide users countermeasures to prevent privacy disclosure according to the different influence weights.

**Keywords:** social network user behavior; information privacy prevention; AHP analysis

## 1. Introduction

With the rapid advancement of technologies, more and more social media networking sites are created and launched such as Facebook, Twitter, Instagram, Weibo, WeChat, etc. They help users stay connected with others, avoid data silos, and provide users the convenience to share their lives and discover their surroundings. It is undeniable that social media networking brings people comfort but also causes lots of privacy issues. Users are aware of the online privacy risks, but still tend to share private information to exchange preferential or personalized services. It shows that people's concern for privacy shows contradictory psychology. The boundary between public space and private space used to distinguish privacy is becoming more and more blurred; we should try our best to balance these two ends, minimizing the risk of privacy disclosure caused by our behavior of transferring privacy for better service, and taking measures to protect important privacy as much as possible [1].

The current research focuses more on people's risk perception and privacy concerns in social media behavior, as well as the preventive measures against privacy disclosure caused by different behaviors. However, there is a lack of classification of social media privacy. Different levels of privacy are of different importance to a person and different behaviors will lead to different categories of privacy disclosure. Therefore, the type of privacy disclosure caused by various behaviors of users on social networks and the degree of privacy disclosure (or protection) need to be quantified.

This study identified four super social network behaviors with twelve sub-behaviors and five types of privacy based on our literature reviews. Our paper studied the impact

of social media behavior on privacy information security and then gives out the privacy prevention level to the users. Finally, protection suggestions are given by professionals.

## 2. Research Framework

### 2.1. Literature Review

2.1.1. Types of User's Behavior

Firstly, we need to be clear about the type of user behavior on social networking.

Some articles analyze and summarize user behavior based on other theoretical research. Ryan and Xenos analyzed social network behavior factors based on five personality traits [2]. They divided Facebook-related behaviors into active social contributions, passive participation, news and information disclosure, and real-time social interaction. At the same time, some scholars believe that user behavior can be researched based on privacy. For example, in combination with mobile social networks' characteristics and based on the research results of the existing index system of privacy disclosure, Wang Xueting and Sun Xiaoya established the index system of privacy disclosure prevention ability through the Expert Investigation Method. They classified user behaviors related to network privacy, privacy concerns, self-disclosure, and privacy protection behaviors. Privacy concerns are expressed fearing that personal information will be leaked when using social platforms. The vast majority of mobile social platform carriers are mobile applications (APP). The following three sub-indicators will be used to measure the privacy concern level of social users: APP background collects user information, APP obtains application rights, and malicious theft. Self-disclosure means social activities are essentially a kind of information transmission behavior. However, the emergence of the internet makes social activities shift to online interaction become traceable. Self-disclosure of social users has become one of the main ways of privacy disclosure including the frequency of sharing your life, the number of times the real information is entered, the degree of self-disclosure to ordinary friends, and self-disclosure to close friends. Privacy protection behavior, because of the concern about privacy, can only reflect netizens' privacy awareness. The final factor related to privacy disclosure is what kind of privacy protection behavior is taken. Behavior can be divided into privacy settings for social applications, traces of social networks, and password management for social networks [3].

2.1.2. Types of Privacy

Several articles have talked about privacy and their different categories. Different articles explain privacy in their ways and divide them into different types.

According to Christian Fuchs [4], privacy is not only about information and communication. He concluded that information and communication is not the only dimension of privacy. Based on his theory, Tavani introduced four types of privacy [5], informational privacy, physical/accessibility privacy, decisional privacy, and psychological/mental privacy. Besides, Christopher Allen advocated another four types of privacy, i.e., defensive privacy, human rights privacy, personal privacy, and contextual privacy. Although not all types are related in social networks, some types can be employed in our concern.

In 2017, Koopsbj et al. did not clearly define privacy's connotation but adopted the typological description method. In a typology of privacy, privacy is divided into eight types: body privacy, spatial privacy, communicative privacy, proprietary privacy, intellectual privacy, decision privacy, associative privacy, and behavior privacy.

Information privacy cannot be regarded as an independent privacy type. Floridi divided privacy into four types: "physical privacy, mental privacy, decisional privacy, and information privacy". Information privacy has independent existence significance. Privacy is freedom from information interference or infringement. With the wide application of networks, social network users are keen on revealing their home address, tourism location, income, and other personal privacy. Traditional privacy, such as spatial privacy and physical privacy, has no longer become the content of privacy.

Gu Liping define that integrated privacy is mainly the data privacy with the arrival of digital era. [6] With the development of biometric information technology, such as fingerprint, face brush, gene technology, etc., intelligent machines can even penetrate the most secret private space by analyzing human facial muscle changes and eye movements, including personal whereabouts, social networks, values, political trends, etc., which are difficult to present biological characteristics, and it is easy to dig them out.

2.1.3. Facebook Case Study

According to statistics from Statista (2019) [7], the number of Facebook daily active users was 1.49 billion in 2018, and the total global number of Facebook users was approximately 1.69 billion in 2020 [8]. The active Facebook users regularly post sensitive data, which can be used to track their activities. However, most users do not know that their posts and updates are in the public domain and can be easily accessed by others. Facebook has a privacy policy statement but it is long and written in jargon which is not easy for users to read and understand [9].

Moreover, there is a risk for third party access control. Facebook cooperates with third parties such that there are lots of third parties' apps on Facebook. The apps can collect users' data and have publishing practices. When there is a conflict between the user's privacy settings and the application's data collection and publishing practices, privacy violation may occur easily. It was found that Facebook's powerful application programming interface (API) enables application developers to collect and publish user data [10].

A study pointed out that the reasons for privacy violations include poor human–computer interaction mechanisms, the static nature of privacy settings, and much work that forces users to maintain their content privacy [11]. Due to our resources constraints, our study is only focused on Facebook users' different behavior and privacy violations.

*2.2. Theoretical Underpinnings*

**Theory 1.** *Social Network Behavior Factors Based on Five Personality Traits.*

Ryan et al. analyzed social network behavior factors based on five personality traits. They divided Facebook-related behaviors into active social contributions, passive participation, news and information disclosure, and real-time social interaction.

**Theory 2.** *Privacy Disclosure Prevention Ability Index System.*

Wang et al. established the index system of privacy disclosure prevention ability through the Expert Investigation Method. They classified the user behaviors related to network privacy concern, privacy protection, self-disclosure, and privacy protection behaviors.

**Theory 3.** *Multi-dimensional Privacy in Online Social Networks Framework.*

Based on the four-dimensional framework proposed by Burgoon et al. [12], Zhang Nan et al. expanded the concept of four-dimensional privacy for social network activities. The four dimensions include the integrity of virtual territory, control of personal factual information, freedom of interaction, and psychological independence. These four types of privacy are the main components of personal privacy in the internet social age [13].

**Theory 4.** *Integrated Privacy Theory.*

Integrated privacy is mainly data privacy with the arrival of digital era. Floridi insisted that data information privacy has independent existence significance. On this basis, Gu Liping also proposed a similar concept. Integrated privacy has two premises: The first premise is that people's words and deeds are digitized to form big data; the second is that big data formed by digitization can be mined with big data technology and become regular information.

### 2.3. Research Model

This above Figure 1 depicts the model based on the analytic hierarchy process. It is divided into four layers. According to the literature review and theoretical underpinnings, we hope to quantify the complex privacy comprehensive score into the score formed by the sum of different privacy scores. In the privacy category layer, we adopt and improve the combination of "*Multi-dimensional Privacy in Online Social Networks Framework*" and "*Integrated Privacy Theory*". These different types of privacy are affected by different social media behaviors. We divided the behaviors on social media into four categories according to "*Privacy Disclosure Prevention Ability Index System*" trying to find out the weight of each behavior affecting each type of privacy. After that, our research team included the 12 most common sub-behaviors into four types of behaviors, and finally wanted to know the weights of these 12 common sub-behaviors in the final privacy score, so as to acquire everyone's privacy prevention score according to everyone's sub-behaviors and put forward behavior suggestions for different people.

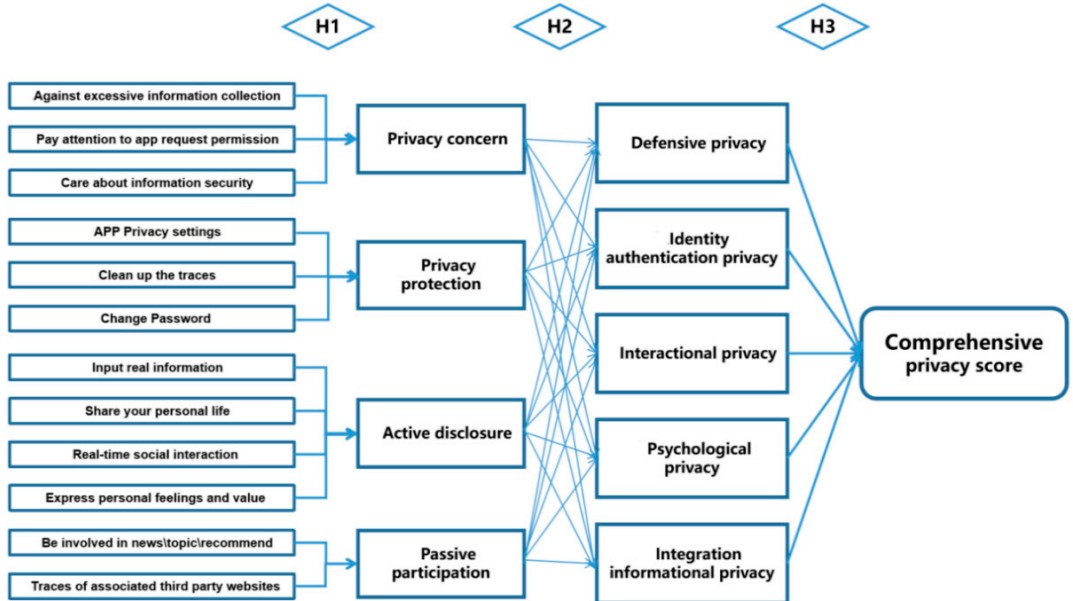

**Figure 1.** Research Model.

The proportion of sub-behavior layer in the corresponding behavior layer is marked as H1; the influence weight of behavior layer on privacy layer is marked as H2; finally, the proportion weight of privacy layer to final privacy score is marked as H3. H1/H2/H3 correspond to the following three hypotheses. In this way, we can quantify each small behavior's impact on the final privacy rating score.

### 2.4. Hypotheses Building

There are three hypotheses in this paper:

**Hypotheses 1 (H1):** *The four types of Internet social behaviors are composed of many specific user sub-behaviors, and sub-behaviors have different weights.*

This hypothesis is based on two theories: "Privacy disclosure prevention ability index system" and "Social network behavior factors based on five personality traits". They proposed a variety of social media internet behaviors; based on them, the criteria level "Type of Behaviors" was set as four items:

1. H2a. Privacy Concern: H1a. Against excessive information collection; H1b. Pay attention to app request permission; H1c. Care about information security.
2. H2b. Privacy Protection: H1d. APP Privacy settings; H1e. Clean up the traces; H1f. Change Password.

3. H2c. Active disclosure: H1g. Input real information; H1h. Share your personal life; H1i. Real-time social interaction. H1j. Express personal feelings and value.

4. H2d. Passive participation: H1k. News and information disclosure; H1l. Traces of associated third party websites.

**Hypotheses 2 (H2):** *Five types of user privacy in the digital age will be lost due to various user behaviors. Each behavior has a different influence weight on specific types of privacy loss.*

Those hypotheses are based on two theories: "Integrated privacy theory" and "Multi-dimensional Privacy in Online Social Networks framework". They put forward a variety of privacy types in the internet era; based on them, we set the decision-making level to five types of privacy:

1. H3a. Defensive privacy—It mainly describes the following aspects of privacy: virtual territory/accessibility.

2. H3b. Identity authentication privacy—It mainly describes the following aspects of privacy: factual/personal/Bodily/Biological.

3. H3c. Interactional privacy—It mainly describes the following aspects of privacy: Communication/comment/share.

4. H3d. Psychological privacy—It mainly describes the following aspects of privacy: emotion/decision/value/knowledge.

5. H3e. Integration informational privacy—It mainly describes the following aspects of privacy: Proprietary/preference/Commercial history and traces.

**Hypotheses 3 (H3):** *The comprehensive privacy score of social media comprises five privacy scores, and the importance of the five kinds of privacy is not the same.*

This layer's weight needs professional knowledge to evaluate, so we adopted the data of the evaluation results of more than 20 experts summarized in the theoretical underpinnings research model "Multi-dimensional Privacy in Online Social Networks framework", whose H3a = 0.193, H3b = 0.339, H3c = 0.121, H3d = 0.220.

We set the final score "Comprehensive privacy score" as the target level. Finally, according to the total score and losing points, this paper provides targeted preventive suggestions to different levels of users.

### 3. Results

*3.1. Descriptive Statistical Analysis*

We targeted teenagers with experience in using social networking to conduct our data collection randomly. The analysis was conducted based on social networking sites. We used the "questionnaire star" statistical software to distribute and recover the questionnaire. After eliminating 5 invalid answers, we collected 416 valid samples and summarize them in the form of Excel. Our online survey was designed with 30 questions. They are all quantitative questions.

The descriptive statistical analysis of the sample is shown in Tables 1 and 2 below.

**Table 1.** Descriptive statistics for twelve sub-behaviors.

| Types of User Behavior | Variables in Twelve User Behaviors | Mean | SD | Description |
|---|---|---|---|---|
| Privacy Concern | Against excessive information collection | 5.67 | 1.43 | The degree of over-collection of information on social media |
| | Pay attention to app request permission | 2.23 | 1.39 | The degree that users pay attention to app request permission and read privacy policy statement |
| | Care about information security | 6.56 | 0.86 | The importance of information security |

**Table 1.** *Cont.*

| Types of User Behavior | Variables in Twelve User Behaviors | Mean | SD | Description |
|---|---|---|---|---|
| Privacy Protection | App privacy settings | 4.04 | 1.52 | The degree of application privacy settings can help reduce privacy leaks |
|  | Clean up the traces | 3.49 | 1.63 | The extent to which clean up the traces on social media can help reduce privacy disclosures |
|  | Change password | 3.32 | 1.71 | The extent to which social media passwords are regularly changed can help reduce privacy disclosures |
| Active Disclosure | Input real information | 5.74 | 1.19 | The extent to which use real information increase privacy disclosures |
|  | Share your personal life | 5.47 | 1.18 | The extent to which share personal life increase privacy disclosures |
|  | Real-time social interaction | 4.72 | 1.55 | The extent to which real-time social interaction has privacy disclosures risk |
|  | Express personal feelings and value | 4.51 | 1.56 | The extent to which express personal feelings and value increase privacy disclosures |
| Passive Participation | Be involved in news/ topic/ recommend | 5.1 | 1.69 | The degree of users that being involved in news/ topic/ recommend |
|  | Traces of associated third party websites | 5.14 | 1.44 | Due to traces of associated third party websites, users received recommended information from social media |

**Table 2.** Descriptive statistics for four behaviors.

| Four Types of User Behaviour | Mean | Description |
|---|---|---|
| Lack of Privacy Concern | 4.82 | The extent to which privacy concern can help reduce privacy disclosures |
| Lack of Privacy Protection | 3.62 | The extent to which privacy protection can help reduce privacy disclosures |
| Active Disclosure | 5.11 | The extent to which use active disclosure increase privacy disclosures |
| Passive Participation | 5.12 | The degree of users that involved in passive participation and that increase privacy disclosures |

*3.2. Analytic Hierarchy Process*

3.2.1. Determine the Measurement Table

This paper used the seven-point system Likert Scale; the specific scoring rules are shown in the Table 3 below.

**Table 3.** Score Likert Scale.

| Score | Means | Score | Means |
|---|---|---|---|
| 1 | The same | / | / |
| 3 | A bit more important | Reciprocal 1/3 | A bit less important |
| 5 | More important | Reciprocal 1/5 | Less important |
| 7 | Extremely important | Reciprocal 1/7 | Extremely unimportant |
| 2, 4, 6, 8 | Median on both sides | Reciprocal $\frac{1}{2}$, $\frac{1}{4}$ | Median on both side |

### 3.2.2. Construct Judgment Matrix

For example, the Figure 2 shows the scores given by experts and the public (the number indicates the importance of the row indicators over the column indicators). The numerical value is obtained after comparing behaviors which reflects the ratio of the importance of the impact on virtual space defensive privacy.

| Defensive Privacy | Privacy Concern | Privacy Protection | active disclosure | passive participation |
|---|---|---|---|---|
| Privacy Concern | 1.00 | 0.26 | 1.15 | 2.43 |
| Privacy Protection | 3.87 | 1.00 | 4.55 | 5.26 |
| active disclosure | 0.87 | 0.22 | 1.00 | 4.59 |
| passive participation | 0.41 | 0.19 | 0.63 | 1.00 |

**Figure 2.** Case of constructing judgment matrix.

Then we can convert the numbers of these importance ratios into a judgment matrix as Figure 3 below.

$$\left\{\begin{array}{llll} 1.00, & 0.26, & 1.15, & 2.43 \\ 3.87, & 1.00, & 4.55, & 5.26 \\ 0.87, & 0.22, & 1.00, & 4.59 \\ 0.41, & 0.19, & 0.63, & 1.00 \end{array}\right\}$$

**Figure 3.** Judgment matrix.

Now we can use SPSS to do the AHP to analyze the judgment matrix and discern the weight of each coefficient. Figures 4 and 5 reflect the influence weight of each behavior on the virtual space defensive privacy from the judgment matrix Figure 3 above.

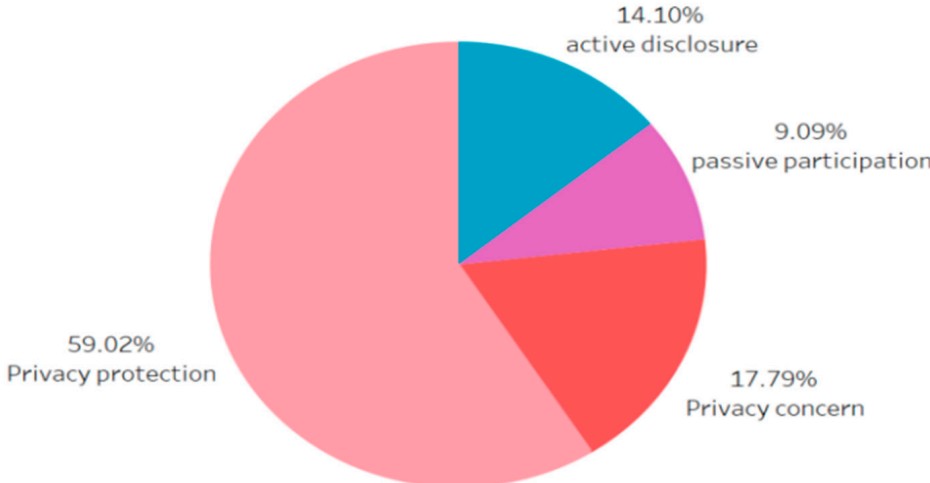

**Figure 4.** Single-level weight graph.

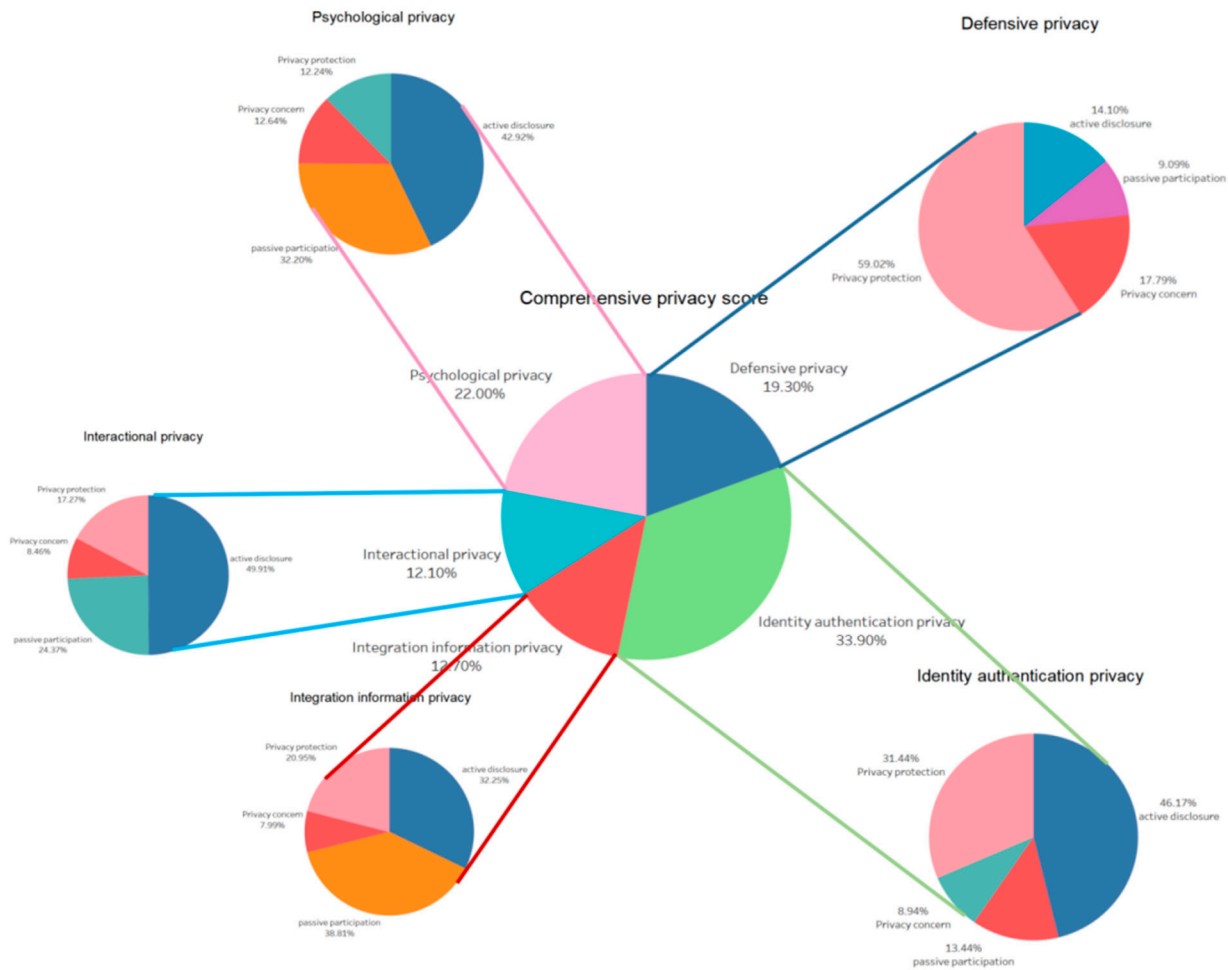

**Figure 5.** Multi-level weight graph.

To understand the relationship weights of four behaviors, five types of privacy and a comprehensive general privacy score, ten judgment matrices are needed. Here we only show the relationship between H2 and H3 layers.

### 3.2.3. Consistency Test

The so-called consistency refers to the logical consistency of judgment thinking. If A is more important than C and B is slightly more important than C, A must be more important than B. This is the logical consistency of judgment thinking; otherwise, there will be contradictions in judgment. Figure 6 below shows the results of the consistency test.

| Consistency test results Summary | | | | |
|---|---|---|---|---|
| Maximum eigenvalue | CI | RI | CR | Consistency test results |
| 4.033 | 0.011 | 0.890 | 0.012 | PASS |

**Figure 6.** Consistency test.

### 3.2.4. Model Analysis Result

The weight of each specific relationship between H1, H2, and H3 is shown in the Figure 7 above. Now we know which behavior has the biggest influence on comprehensive privacy level. We can also use this set of functions to evaluate each social network user's privacy prevention level.

| Privacy Classification | Internet Behavior | Behavior Weights on Privacy | Related Behavior | Weights on Internet Behavior | Weights on Privacy | Weights on Total score |
|---|---|---|---|---|---|---|
| H3a: 0.193 | Privacy Concern | H2a: 0.1779 | H1a | 0.394 | 0.070 | 0.014 |
| | | | H1b | 0.216 | 0.038 | 0.007 |
| | | | H1c | 0.390 | 0.069 | 0.013 |
| | Privacy Protection | H2b: 0.5902 | H1d | 0.374 | 0.221 | 0.043 |
| | | | H1e | 0.323 | 0.191 | 0.037 |
| | | | H1f | 0.303 | 0.179 | 0.034 |
| | Active Disclosure | H2c: 0.141 | H1g | 0.282 | 0.040 | 0.008 |
| | | | H1h | 0.246 | 0.035 | 0.007 |
| | | | H1i | 0.237 | 0.033 | 0.006 |
| | | | H1j | 0.235 | 0.033 | 0.006 |
| | Passive Participation | H2d: 0.0909 | H1k | 0.488 | 0.044 | 0.009 |
| | | | H1l | 0.512 | 0.047 | 0.009 |
| H3b: 0.339 | Privacy Concern | H2a: 0.0894 | H1a | 0.394 | 0.035 | 0.012 |
| | | | H1b | 0.216 | 0.019 | 0.007 |
| | | | H1c | 0.390 | 0.035 | 0.012 |
| | Privacy Protection | H2b: 0.3144 | H1d | 0.374 | 0.118 | 0.040 |
| | | | H1e | 0.323 | 0.102 | 0.034 |
| | | | H1f | 0.303 | 0.095 | 0.032 |
| | Active Disclosure | H2c: 0.4617 | H1g | 0.282 | 0.130 | 0.044 |
| | | | H1h | 0.246 | 0.114 | 0.039 |
| | | | H1i | 0.237 | 0.110 | 0.037 |
| | | | H1j | 0.235 | 0.108 | 0.037 |
| | Passive Participation | H2d: 0.1344 | H1k | 0.488 | 0.066 | 0.022 |
| | | | H1l | 0.512 | 0.069 | 0.023 |
| H3c: 0.121 | Privacy Concern | H2a: 0.0846 | H1a | 0.394 | 0.033 | 0.004 |
| | | | H1b | 0.216 | 0.018 | 0.002 |
| | | | H1c | 0.390 | 0.033 | 0.004 |
| | Privacy Protection | H2b: 0.1727 | H1d | 0.374 | 0.065 | 0.008 |
| | | | H1e | 0.323 | 0.056 | 0.007 |
| | | | H1f | 0.303 | 0.052 | 0.006 |
| | Active Disclosure | H2c: 0.4991 | H1g | 0.282 | 0.141 | 0.017 |
| | | | H1h | 0.246 | 0.123 | 0.015 |
| | | | H1i | 0.237 | 0.118 | 0.014 |
| | | | H1j | 0.235 | 0.117 | 0.014 |
| | Passive Participation | H2d: 0.2437 | H1k | 0.488 | 0.119 | 0.014 |
| | | | H1l | 0.512 | 0.125 | 0.015 |
| H3d: 0.220 | Privacy Concern | H2a: 0.1264 | H1a | 0.394 | 0.050 | 0.011 |
| | | | H1b | 0.216 | 0.027 | 0.006 |
| | | | H1c | 0.390 | 0.049 | 0.011 |
| | Privacy Protection | H2b: 0.1224 | H1d | 0.374 | 0.046 | 0.010 |
| | | | H1e | 0.323 | 0.040 | 0.009 |
| | | | H1f | 0.303 | 0.037 | 0.008 |
| | Active Disclosure | H2c: 0.4292 | H1g | 0.282 | 0.121 | 0.027 |
| | | | H1h | 0.246 | 0.106 | 0.023 |
| | | | H1i | 0.237 | 0.102 | 0.022 |
| | | | H1j | 0.235 | 0.101 | 0.022 |
| | Passive Participation | H2d: 0.322 | H1k | 0.488 | 0.157 | 0.035 |
| | | | H1l | 0.512 | 0.165 | 0.036 |
| H3e: 0.127 | Privacy Concern | H2a: 0.0799 | H1a | 0.394 | 0.032 | 0.004 |
| | | | H1b | 0.216 | 0.017 | 0.002 |
| | | | H1c | 0.390 | 0.031 | 0.004 |
| | Privacy Protection | H2b: 0.2095 | H1d | 0.374 | 0.078 | 0.010 |
| | | | H1e | 0.323 | 0.068 | 0.009 |
| | | | H1f | 0.303 | 0.063 | 0.008 |
| | Active Disclosure | H2c: 0.3225 | H1g | 0.282 | 0.091 | 0.012 |
| | | | H1h | 0.246 | 0.079 | 0.010 |
| | | | H1i | 0.237 | 0.076 | 0.010 |
| | | | H1j | 0.235 | 0.076 | 0.010 |
| | Passive Participation | H2d: 0.3881 | H1k | 0.488 | 0.190 | 0.024 |
| | | | H1l | 0.512 | 0.199 | 0.025 |

**Figure 7.** Model analysis result.

*3.3. User Behavior Empirical Analysis*

3.3.1. User Behavior Evaluation Model

In the index system of personal privacy disclosure prevention interval, the weights of each code of conduct layer's indexes to the privacy target layer are established, and the scores collected from the questionnaire are combined to make a comprehensive evaluation. The weighted calculation of social users' grading system for each indicator is processed into a score (out of 100).

$$Z_i = \left( \sum_{i=1}^{12} W_i X_i \right) * (100/F)$$

In the above formula, *Wi* represents the weight of the behavior criteria interval index relative to the privacy target interval in the 'ith' question of the questionnaire, *Xi* represents the score of the 'ith' question, *F* represents the full score of the questionnaire. *Z* represents the final score of the user's questionnaire (out of 100). The maximum value of each question is 7 points, and the minimum value is 1 point. After the full score of 100 is divided into

10 segments, the following privacy protection intervals are obtained, as shown in the Table 4 below.

**Table 4.** Comparison table of privacy protection interval and grade.

| Grade | 0–10 | 10–20 | 20–30 | 30–40 | 40–50 | 50–60 | 60–70 | 70–80 | 80–90 | 90–100 |
|---|---|---|---|---|---|---|---|---|---|---|
| Interval | 1 | 2 | 3 | 4 | 5 | 6 | 7 | 8 | 9 | 10 |

The score of each question in each investigator's questionnaire is multiplied by the corresponding weight value, and then the score is calculated and divided into the corresponding score interval. After that, the histogram of the score distribution of the privacy disclosure prevention status of 416 users is obtained, as shown in the following Figure 8.

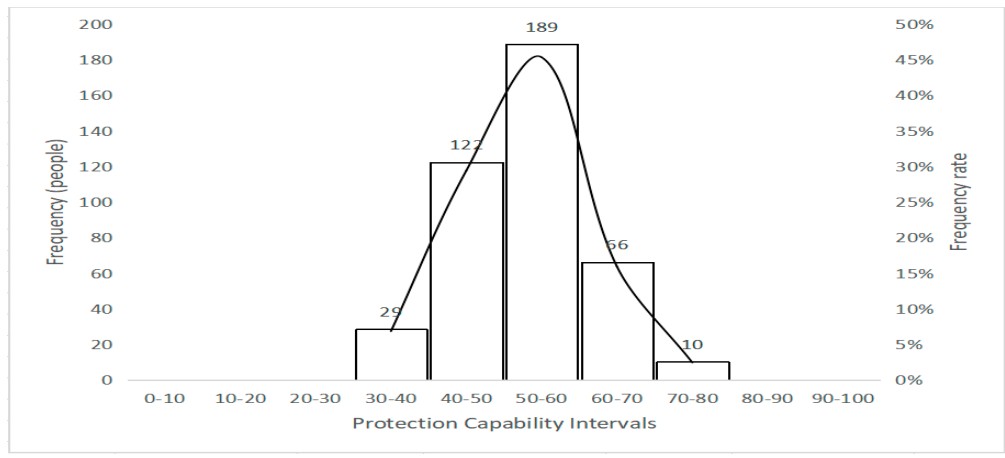

**Figure 8.** Frequency histogram of privacy protection capability.

### 3.3.2. Survey Result Analysis

According to the survey results, we divided the 416 samples into different levels based on average marks for analysis. Additionally, we found that there were significant differences among levels in some subsets.

In Figure 9, there are five defined intervals. Since all data samples only stay at 30–80 points, for convenience, we can artificially name these five intervals as A–E level from grade high to low.

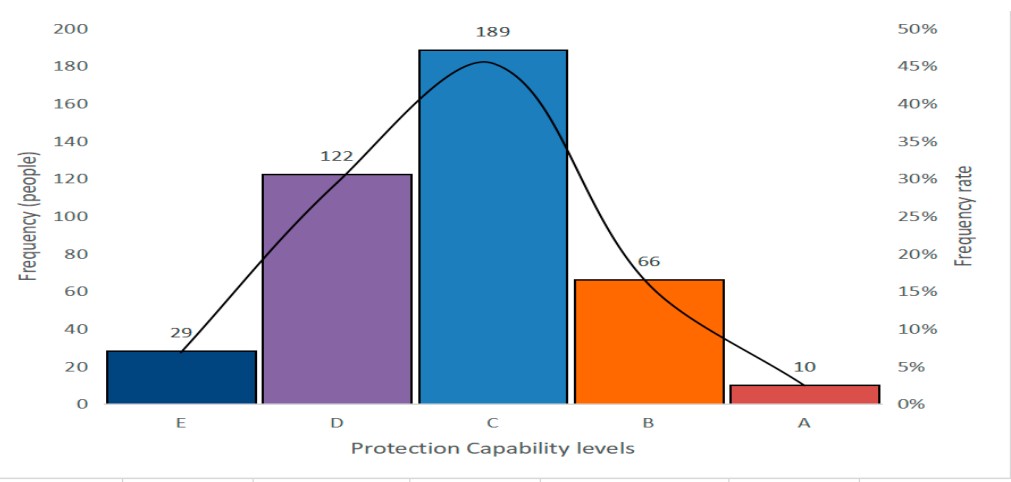

**Figure 9.** Frequency histogram of privacy protection capability (A–E).

The samples of level E were the lowest average marks in the result. It means that the respondents in level E lack privacy awareness compared with other levels. The lowest rating was the changing password of privacy protection. It shows that users do not change their password periodically. It may become a risk of their private information, which increases the chance of cracking the password by hackers to steal private information.

The lowest rating of level D was the news and information disclosure of passive participation. It reveals that these respondents often receive advertisements or news on social media. It is possible that there may have leakage of their personal information.

Level C contained the largest number of samples in the result. The lowest rating of level C was the care about information security of privacy concerns. It shows that the majority of people normally do not concern themselves about privacy much.

The respondents of level B were above the average in the result. They are defined as having an awareness of privacy protection. The lowest rating of level B was the pay attention to the app request permission of privacy concern. It shows that most people do not care about the permissions that app asks for from users, which has also become the sociological reason why app excessively collects information. If there is a malicious app and it is granted permission, the risk of leaking privacy information is increased.

The sample of Level A is very interesting. They have a certain level of privacy prevention in all aspects of their behavior. However, they still think they "don't have privacy awareness". It can be seen that these groups are highly sensitive to privacy. Maybe these people are experts in the field of data and will think deeply about how to better protect privacy. It is precisely because they know that there is still much room to improve the protection of privacy that they feel that they have not done enough.

From Figure 10 above, there was a significant difference between level E to level A in the clean-up the traces of privacy protection. The rating of level E was only 34.3% but the rating of level A was 100%. Even level C was 57.5%. People who do not focus on privacy protection do not clean up their history often. On the other hand, people who are aware of privacy protection clean up their traces periodically.

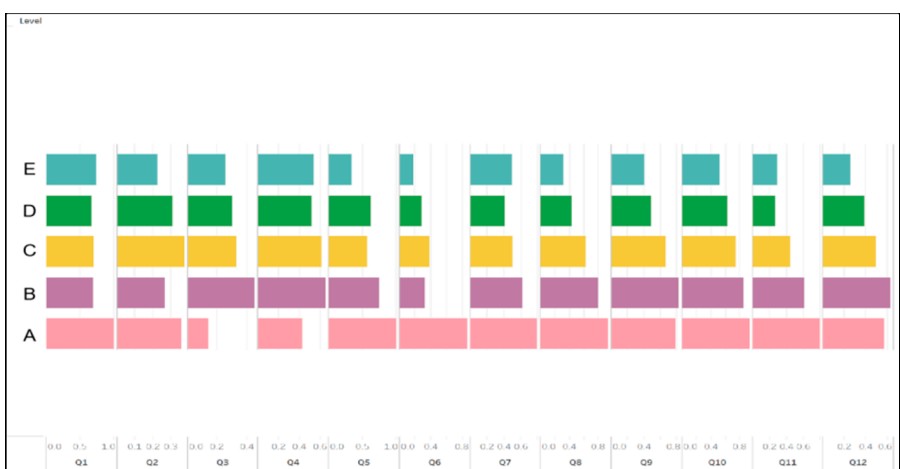

**Figure 10.** Questionnaire—user behavior 5 intervals.

The result of the changing password of privacy protection shows that there is a significant difference. The lowest rating was level E, which was 17.1%; the highest rating was 85.7% from level A. The second-highest rating was 38.2% from level C. The difference between level E and level A was large. It shows that people do not often change their passwords, even though for level C, their rating was lower than 50%.

Another significant difference is the shared personal life of active disclosure. The lowest rating was 31.4% from level D. The highest rating was 92.9% from level A. The second-highest rating was 79.1% from level B. It reveals that people who are not aware of privacy security often share their personal life on social media. Those shared contents could be photos, locations, or habits, etc. These all are privacy data and are easily taken by

others in social media. People who care about privacy are more likely to set up rights to control information visibility.

## 4. Discussion

The above descriptive statistical table shows that respondents think passive participation and active disclosure are the riskiest in privacy disclosure.

## 5. Conclusions

After we learned that 160,000 Facebook pages were hacked every day [14] and that Facebook faced a $5 billion fine for privacy violations [15], we began to carry out this research. Trying to make social media users aware of the relationship between behavior and privacy.

We quantified the degree of different privacy disclosure behind each user's behavior. In this regard, we need to pay more attention to those behaviors with greater weight. Based on the theoretical model and data collection of experts and users, two conclusions were found.

First, from the comparison of importance and the conclusion of AHP is that: among the privacy types that affect the comprehensive score of privacy, the order is identity authentication privacy (0.339), psychological privacy (0.220), virtual space defensive privacy (0.193), integrated information privacy (0.127), and interactive privacy (0.121).

Specifically, for each type of privacy, such as virtual space defensive privacy, the order of influence of each behavior on it is: privacy protection behavior (0.5902), privacy concern behavior (0.1779), active interaction behavior (0.141), and passive participation behavior (0.0909).

For a specific type of behavior, the weight of sub-behavior can be ranked as follows. For example, the order of privacy concern behavior is: worry about over-collecting of information (0.394), degree of information security concern (0.390), and concern about app access rights (0.216).

With these weights, we can rate the behavior of users, so we have a series of second conclusions.

The lowest rating of all levels shows that people normally do not change their password periodically, even those who have some awareness of privacy protection. Nowadays, most social networks provide a feature which is called 'Keep login'. People use this function on mobile devices frequently. When they enter the social network, they do not need to enter a password. Therefore, they may think that the password may not be important and changing the password is unnecessary. However, we suggest people change their passwords periodically, say, every half year. To achieve this purpose, people should set a timer to alert themselves. Some companies require their employees to update the passwords of the ID accounts after a period. People who are in those companies can update the passwords of social media accounts at the same time. They should avoid password reuse at all costs when updating the passwords. Although people may think it is annoying, many social media accounts are being hacked every day.

Moreover, the lowest rating reveals that people normally do not pay attention to the request for apps' permission. Many apps require permissions of some aspects, such as photos, contacts, on mobile devices. Some of them must have permission; otherwise, users cannot use the apps. When people use these apps, they have to accept the requests, even paying attention to the requests. If they keep using these types of apps, they ignore the request for permission over time.

The conclusion of empirical analysis of behavior tells us that most people normally are not aware of their privacy. For instance, most people often share their personal life on social media. Although it is the main activity on social media, it still raises a problem with the disclosure. This disclosure is activated by people's behavior. It is impossible to restrict people from sharing but people should be conscious about what they share and who can view the contents. People also should limit other people who could view the

shared contents. Usually, there is a setting of who can view the posts on social media. People should set it to those they know.

This study provides the weight of various behaviors that affect privacy and evaluates users' level of privacy prevention. In this way, we can provide specific suggestions for different groups of people and improve the privacy prevention of the whole people from the users' perspective. Additionally, this study aimed to enrich our understanding of people's privacy and adjust their behavior in social networks. Privacy protections are crucial for both people and social networks. While the discussion proposed in this study is preliminary, it shows the value of adding specificity about privacy.

## 6. Limitations and Improvements

Our research team only provides some references on the relationship between social media behavior and personal privacy protection based on student samples and thinking. Whether the questionnaire design can significantly reflect the facts needs to be further optimized, and we hope to promote larger-scale research on data security and personal privacy in future when we have sufficient resources.

**Author Contributions:** Conceptualization, Y.L.; methodology, Y.L.; software, Y.L. and P.Y.K.; Data analysis, Y.L. and P.Y.K.; validation, P.Y.K.; formal analysis, P.Y.K. and Y.H.C.; investigation, Y.H.C.; writing—review and editing, Y.L., P.Y.K. and W.K.T.; supervision, W.K.T. All authors have read and agreed to the published version of the manuscript.

**Funding:** This research received no external funding.

**Institutional Review Board Statement:** Ethical review and approval were waived for this study, due to the fact we conducted the questionnaire survey in Hong Kong and strictly abided by the Personal Data (Privacy) Ordinance (PCPD) in the Data Privacy Law of HKSAR through making a declamation at the start of our questionnaire survey, which is 'This survey is an anonymous survey, only for our research study this time, you are not compulsory to participate in this survey. This questionnaire survey research is entirely dependent on your voluntary help, and all your information will be kept strictly confidential and will be cleared after the survey. If you can take a few minutes to participate in this survey, we would be very grateful!'

**Informed Consent Statement:** Informed consent was obtained from all subjects involved in the study.

**Data Availability Statement:** Not applicable.

**Conflicts of Interest:** The authors declare no conflict of interest.

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
