# Peer review of "Impact of Social Media Behavior on Privacy Information Security Based on Analytic Hierarchy Process"

_information, doi:10.3390/info13060280_

Round 1
Reviewer 1 Report
Originality
Research contribution in the paper can be identified; this study can be justified as innovative.
Title
The title is correct as it reflects the objective of the work. The title is too long.
Abstract
The abstract needs to be rewritten, with a structured summary of the evaluation of studies, conclusions and implications of key findings, and so on. For, instance: 1) Place the background question asked in a broad context and highlight the purpose of the study; (2) Briefly describe the main methods or treatments used; (3) Summarize the main findings of the article; (4) Indicate the main conclusions or interpretations.
Keywords
The keywords are adequate.
Introduction
The authors should better highlight the innovative aspects of their work in the manuscript.
The introduction part is not adequate. The Introduction should consist of background about the topic being studied, i.e. the rationale for undertaking this study for filling a research gap.
The introduction part should be short, and highlights the significance of the study.
A research gap should be emphasized why the topic is selected. In the Introduction, answer two main questions: "Why was this particular study needed to fill this gap in scientific knowledge that currently exists?" Describe the rationale in the context of what is known and not in this research area. The problem statement based on the previous results and the novelty of this study should also be declared here.
Literature review
This section includes many relevant references and the authors provide solid theoretical foundations for the analysis using appropriate references.
Methodolgy
Indicate limits and advantages of methods. Indicate alternative methods.
Results
Obtained results are developed based on methodological justification, they are supported by relevant presentation. Please, edit every figures and tables. Enlarge table and figure according to the formal template.
Conclusion and implications
The conclusion section should highlight more clearly how the results compare with the recent literature review.
In the conclusion of the paper, it is necessary to connect obtained results with the main purpose of the research and the main hypothesis/research questions.
What are the main constraints of your research that need to be considered when interpreting the results obtained and making conclusions?
Make sure that all figures and tables are cited within the text and are cited in consecutive order.
Author Response
Dear Reviewer 1:
Thank you for your comments and suggestions. The following is my reply and revision:
Suggestion1 - Title: The title is too long.
I changed the title more accurately to “Impact of social media behavior on privacy information security based on analytic hierarchy process”
Suggestion2 - Abstract: The abstract needs to be rewritten, with a structured summary of the evaluation of studies, conclusions and implications of key findings, and so on. For, instance:
1) Place the background question asked in a broad context and highlight the purpose of the study; 2) Briefly describe the main methods or treatments used;
3) Summarize the main findings of the article;
4) Indicate the main conclusions or interpretations.
Thank you for your very pertinent advice. I revised a new edition of the abstract. The new abstract highlights and refines the purpose, methods, process and conclusion significance.
Abstract “In the era of global social media, Internet users' privacy rights have been weakened, and the insight and alertness of individuals for privacy disclosure are decreasing. The security and flexibility of the system are usually the two ends of the measurement standard. While more and more users pursue the intelligence and convenience of using social media applications, letting big data technology and AI algorithms learn and use our private data cannot be avoided. On the basis of literature review, this paper summarizes four categories of social media user behavior, which are divided into privacy concern behavior, privacy protection behavior, active disclosure behavior and passive participation behavior. Using analytic hierarchy process, this paper explores their relationship with five different types of privacy: defensive privacy, identity authentication privacy, interactive privacy, psychological privacy and integration information privacy, We finally got the most common twelve kinds of sub Internet user behavior on the degree of personal privacy disclosure, and give users countermeasures to prevent privacy disclosure according to the different influence weights.”
Suggestion3 - Introduction: The authors should better highlight the innovative aspects of their work in the manuscript. The introduction part is not adequate. The Introduction should consist of background about the topic being studied, i.e. the rationale for undertaking this study for filling a research gap. The introduction part should be short, and highlights the significance of the study. A research gap should be emphasized why the topic is selected. In the Introduction, answer two main questions: "Why was this particular study needed to fill this gap in scientific knowledge that currently exists?" Describe the rationale in the context of what is known and not in this research area. The problem statement based on the previous results and the novelty of this study should also be declared here.
Thank you for your advice. My introduction did not clearly explain the significance and innovation of the research before, so I highlight the innovative aspects of our work in the manuscript. Combined with the previous research direction, focusing on explaining this question "Why was this particular study needed to fill this gap in scientific knowledge that currently exists?". I have changed the introduction as below:
With the rapid advancement of technologies, more and more social media networking sites are created and launched such as Facebook, Twitter, Instagram, Weibo, etc. It helps stay connected with others, avoid data silos, and gives users the convenience to share their lives and discover their surroundings. It is undeniable that social media networking brings people comfort but also causes lots of privacy issues. Users start to be aware of the online privacy risks, but still tend to share private information to exchange preferential or personalized services. It shows that people's concern for privacy shows contradictory psychology. The boundary between public space and private space used to distinguish privacy is becoming more and more blurred, we should try our best to balance these two ends, minimize the risk of privacy disclosure caused by our behavior of transferring privacy for better service, and take measures to protect important privacy as much as possible.
The current research focuses more on people's risk perception and privacy concerns in social media behavior, as well as the preventive measures against privacy disclosure caused by different behaviors. However, there is a lack of classification of social media privacy. Different privacy is of different importance to a person, and different behaviors will lead to different categories of privacy disclosure. Therefore, the type of privacy disclosure caused by various behaviors of users on social networks and the degree of privacy disclosure (or protection) need to be quantified.
This study identifies four super social network user behavior with twelve sub behavior and five type of privacy based on our literature reviews. Our paper studies the impact of social media behavior on privacy information security, and then give out the privacy prevention level to the users. Finally, protection suggestions are given by professionals
Suggestion4 - Conclusion and implications. The conclusion section should highlight more clearly how the results compare with the recent literature review. In the conclusion of the paper, it is necessary to connect obtained results with the main purpose of the research and the main hypothesis/research questions. What are the main constraints of your research that need to be considered when interpreting the results obtained and making conclusions? Make sure that all figures and tables are cited within the text and are cited in consecutive order.
Enlarged table and figure according to the formal template. ✔
how the results compare with the recent literature review - We refer to the models and concepts in the literature review, but our model is self-created so the data are not comparable.
Conclusion is necessary to connect obtained results with the main purpose of the research and the main hypothesis/research questions. ✔ Our purpose is to quantify the degree of different privacy disclosure behind each user's behavior so that we can know which behaviors users should improve and which are more important.
Limitations and improvements. ✔
all figures and tables are cited within the text and are cited in consecutive order. ✔
Reviewer 2 Report
The authors raise a very interesting topic related to sharing privacy in social networks or, more broadly speaking, digital hygiene. This is especially important due to the unconscious and at the same time very massive use of social networks by the general public. At the same time, it is obvious that failure to maintain proper digital hygiene can lead to the loss of property or even identity.
The layout of the authors' work is interesting, because in my opinion, apart from a clear introduction, they show the theoretical foundations in the right order by selecting the appropriate sources. This gives the reader a proper reference to this research base.
The authors presented very interesting hypotheses in their work, which further on the basis of the conducted research and statistical analysis, refer to them accordingly. It is the proper discussion, next to the tables with the results presented earlier, that is the undoubted advantage of the work here. What is more, the conclusions provide grounds for use also for people who are not experienced in the subject of research. The applications presented by the authors have a wide applicability due to the need to promote proper digital hygiene.
The article is very interesting and worth publishing.
Adding more gaps between figures and tables and text because now it blends together and it is difficult to analyze the content.
Things like the pattern on page 10 should be numbered and in the text then a reference to that pattern.
Tables should be centered, e.g. table 5 is not.
There is a problem with table 1. The subtitles in the attachment screen overlap. Maybe it's my reader's issue, but it's better to check and possibly correct it. (screen attached)
After these corrections, I consider it is ready for publication.

Author Response
Dear Reviewer 2:
Thank you for your recognition of this article. The following is my reply and revision.
Suggestion1: Adding more gaps between figures and tables and text because now it blends together and it is difficult to analyze the content.
I try to increase the gap between the charts as much as possible to make the charts clearer.
Suggestion2: Things like the pattern on page 10 should be numbered and in the text then a reference to that pattern.
It is a graphics imported from our other Word document. Since it has so many items inside, we used another Word to make this table and then fine-tune and fit the size when we imported it into this main document.
Suggestion3: Tables should be centered, e.g. table 5 is not.
Table 5 is a little redundant, so I deleted it
Suggestion4: There is a problem with table 1. The subtitles in the attachment screen overlap. Maybe it's my reader's issue, but it's better to check and possibly correct it. (screen attached)
Maybe it's the issue of “MS Word” version. In fact, I can't see the error as shown in the screen.
Round 2
Reviewer 1 Report
I accept this manuscript.